



# Brief communication: Reanalyses underperform in cold regions, raising concerns for climate services and research

Bin Cao[1] and Stephan Gruber[2]

[1]State Key Laboratory of Tibetan Plateau Earth System, Environment and Resources (TPESER), National Tibetan Plateau Data Center, Institute of Tibetan Plateau Research, Chinese Academy of Sciences, Beijing, China
[2]Department of Geography and Environmental Studies, Carleton University, Ottawa, Ontario, Canada

**Correspondence:** Bin Cao (bin.cao@itpcas.ac.cn)

**Abstract.** Many changes in cold regions are amplified by nonlinear processes involving ice, and have important consequences locally and globally. We show that the average ensemble spread of the mean annual air temperature (1.5 °C) in the reanalyses is 90% greater in cold regions compared to the other regions and shows pronounced disagreement in the trend. The ensemble spread in the mean annual maximum snow water equivalent is found greater than the ensemble mean. The reduced quality
of reanalyses in cold regions, coinciding with sparse in situ observations and low population, points to challenges in how we represent cold-regions phenomena in simulation systems and limits our ability to support climate research and services.

## 1 Introduction

Cold regions are experiencing the planet's strongest warming (Masson-Delmotte et al., 2021). This has local consequences for ecosystems and people, as well as global consequences in affecting weather patterns elsewhere through teleconnections
(Overland et al., 2016; Cohen et al., 2021), and through feedbacks and potential tipping elements (Richardson et al., 2023) affecting climate. Despite their sparse population and remoteness, cold regions disproportionately affect climate change and its consequences everywhere (Abram et al., 2019).

Understanding cold regions is important for informing local climate-change adaptation and climate action globally. Their climate conditions and dynamics, however, can be subject to disagreement (e.g., Thorne, 2008; Graversen et al., 2008; Gao
et al., 2018). As many cold-region processes react nonlinearly to changes near 0 °C due to the ice-water phase transition, their analysis and simulation are extra sensitive to errors. In addition to these challenges, sparse in-situ observations increase the need for atmospheric reanalyses as a tool for supporting climate research and services.

Because observations are sparse, the quality of reanalyses is expected to be lower in cold regions. For example, previous studies suggested that the climate signal in cold regions could be different depending on the datasets used (Huang et al.,
2017; Wang et al., 2017) and concluded that the 'warming hiatus' in the Arctic may be an artifact. Other studies report the performance of reanalyses for specific variables and places (e.g., Graham et al., 2019; Cao et al., 2020). While reanalyses are of higher importance in cold regions, their quality is also less well known than elsewhere.

This study uses a simple and intuitive analysis to illustrate and contextualize a critical gap in knowledge and capabilities for representing and analyzing the Earth system. We quantify the relative quality of five state-of-the-art reanalyses in cold regions





to inform the application of reanalysis products and to motivate further improvements specific to cold environments. We use the average ensemble spread (e.g., Fortin et al., 2014) as an observation-independent measure of relative quality. We focus on the mean annual air temperature (MAAT) and maximum snow water equivalent (maxSWE) because of their dominant control over cold environments and their intuitive interpretation.

## 2 Materials and Methods

### 2.1 Reanalyses

Five state-of-the-art reanalyses, JRA-3Q, ERA5, MERRA-2, JRA-55, and NCEP2 are investigated (Table 1). The 10-member ensemble of ERA5, which quantifies uncertainties in the ERA5 assimilation and modeling system, was also included here to show how only parameter uncertainty in one reanalysis system compares with the differences between reanalyses. While JRA-3Q, ERA5, and JRA-55 are produced using the most advanced four-dimensional variational (4DVAR) assimilation, NCEP2

and MERRA-2 use the three-dimensional variational method (3DVAR). As better performance is expected from the newer 4DVAR reanalyses, we also analysed them separately. The decades 1991–2020 were used, likely a period of high quality for reanalyses.

The in situ observations and population information are from the (CDS) and for International Earth Science Information Network CIESIN Columbia University (2018), respectively, representing conditions in 2020. Population information is not

available for the Antarctica, and no population is assumed for these regions.

### 2.2 Near-surface air temperature downscaling

The ERA5 grid has the highest spatial resolution (0.25°) and was used as the common grid for this analysis. We conducted a 3-D downscaling on the other reanalyses to produce the surface air temperature at ground elevation with a consistent spatial resolution by adopting the algorithms present by Cao et al. (2017, 2019). Surface and upper air (pressure level) temperatures are

first regridded to the ERA5 grid with 2D linear interpolation. Then, the 2D interpolated surface air temperature was refined by adding the lapse rate derived from a linear extrapolation of the lowest two pressure levels above the ground. If an inversion was present, a zero lapse rate was used. The downscaling algorithm significantly removes the influences of inconsistent resolution and improve the data comparability (Figure S1).

### 2.3 Snow water equivalent

The snow water equivalent (SWE) is derived by multiplying snow density and snow depth where it is not available as a variable directly. Only areas with a mean maximum snow depth during 1991–2020 greater than 0.05 m (assuming that the snow density is 250 kg m$^{-3}$) are shown. In contrast to the other reanalyses, MERRA-2 contains a precipitation correction based on observations (Reichle et al., 2017). The corrections were implemented in the coupled model, but did not extend to latitudes north of 62.5° N.





## 2.4 Reanalyses ensemble

Most available high-quality observations are assimilated by reanalyses, therefore, stand-alone assessment is challenging. Indeed, reanalyses are produced via complex systems, and they generally differ in observation system, assimilation system, processing algorithm, and employed physical laws, identifying one or more reliable datasets is a difficult matter (Xu and Powell, 2010). For these reasons, average ensemble spread, is widely used in Earth system research to infer the reliability of an ensemble prediction system (Fortin et al., 2014). We used the average ensemble spread ($ens_s$) as an intuitive measure of how different the target variable could turn out based on an arbitrary choice of one of multiple well regarded reanalyses. The metric $ens_s$, therefore, indicates where reanalyses, taken together as representing our ability to quantify atmospheric state, are more and where they are less accurate. With $V_m = [m_1, m_2, ..., m_n]$ being a set of target variables from different reanalyses ($m$), the average ensemble spread of a variable $V$ is given as

$$V_s = \sqrt{\left(\frac{n+1}{n}\right) s^2} \, , \tag{1}$$

where $n$ is the ensemble size, $s^2$ is the unbiased estimator for the variance of the ensemble members, and is given as

$$s^2 = \left(\frac{1}{n-1}\right) \sum_{i=1}^{n} (\overline{V_m} - m_i)^2 \, , \tag{2}$$

where $\overline{V_m}$ is the ensemble mean of the target variable. The variables ranges $V_s$ used here are the mean annual air temperature ($MAAT_s$) and the maximum snow water equivalent ($maxSWE_s$). The relative average ensemble spread, the ensemble spread divided by the ensemble mean, was used for the maxSWE because of their strong variability.

## 2.5 Cryosphere occurrence

The fractional occurrence of glaciers, ice sheets, snow cover, permafrost, and seasonally frozen ground are used to illustrate the cryosphere context of our findings. Glaciers are from the Randolph Glacier Inventory (Consortium, 2023), snow-cover extent

**Table 1.** The five state-of-the-art reanalyses used in this study.

| Reanalysis | Resolution [°] | Assimilation | References |
|---|---|---|---|
| JRA-3Q | 0.375 | 4DVAR | Harada et al. (2021) |
| ERA5 | 0.25 | 4DVAR | Hersbach et al. (2020) |
| ERA5-ENS[a] | 0.50 | 4DVAR | Hersbach et al. (2020) |
| MERRA-2 | 0.5×0.625 | 3DVAR | Gelaro et al. (2017) |
| JRA-55 | 0.5625 | 4DVAR | Kobayashi et al. (2015) |
| NCEP2 | 2.5 | 3DVAR | Kanamitsu et al. (2002) |

[a]ERA5-ENS refers to the 10-member ensemble of ERA5.



is from Estilow et al. (2015), seasonally frozen ground is from Kim et al. (2017). Since snow and seasonal frozen ground have
a spatial and a temporal extent, their occurrence is derived as the presence probability for the analysis period of 1991–2020.

The permafrost zonation index (PZI) is used to derive permafrost extent (Gruber, 2012), and scaled to 0–100%. The heuristic-
empirical model links the extent of permafrost with the long-term MAAT.

$$PZI = \frac{1}{2}erfc(\frac{MAAT + \mu}{\sqrt{2\sigma^2}}) \times 100 \tag{3}$$

where $\mu = 4.8$ and $\sigma = 2.54$ are model parameters from Gruber (2012). The MAAT is derived from the ensemble mean of all
five reanalyses from 1991 to 2020. Even though the parameters have been calibrated for the period 1961–1990, we use them
unchanged in a warmer period (i.e., 1991–2020), were slightly more permafrost than indicated may therefore persist in the
subsurface.

For MAAT bins of 0.1 °C, the mean covered area is derived as the covered area by specific cryosphere element divided by
the total terrestrial areas including the ice sheets. Note that grid cells are weighted by area before further analyses.

## 3 Results

### 3.1 Lower agreement among reanalyses in cold regions

We consider cold regions to be terrestrial areas, including the Greenland and Antarctic ice sheets, with a MAAT below 0 °C.
The average ensemble spread of MAAT ($MAAT_s$) is higher in cold regions (Fig. 1). Using all five reanalyses ($MAAT_s^{all}$), it is
about 1.5 °C (0.5–3.0 °C, hereafter, values are reported as mean, 10th to 90th percentile) in cold regions, or about 90% higher
than in other regions (0.8, 0.3–1.5 °C). The MAAT trend shows a similar pattern, the spread of the mean warming trend in cold
regions is about 60% higher than that in other terrestrial areas, i.e., 0.24 °C dec$^{-1}$ (0.10–0.42) $vs.$ 0.15 °C dec$^{-1}$ (0.06–0.25),
and is about 56% of the ensemble mean.

Because maxSWE has strong spatial variability, we report its relative spread (Fig. 2C & D). Its high average (105, 51–206%)
shows that, on average, the variation of maxSWE between reanalyses is greater than their ensemble mean. This suggests that
reanalyses face challenges in supporting services and research related to snow. We found the greatest $maxSWE_s$ occur in
mountain regions, for example, the high Asia Mountains (Fig. 2C & D), where snowmelt water is an essential supply to
millions of people downstream for irrigation, hydroelectric power, and consumption Qin et al. (2020); Kraaijenbrink et al.
(2021). Previous studies reported that the snow uncertainties in mountains are related to the performance of numerical weather
prediction models in representing precipitation and snow processes Cao et al. (2020); Domine et al. (2019), especially the
well-known bias in MERRA-2 precipitation Reichle et al. (2017).

Compared to all five reanalyses, the 4DVar reanalyses show a reduced spread in MAAT (1.3, 0.3–2.9 °C) and its trend (0.13,
0.04–0.24 °C dec$^{-1}$), as expected from a consistent and more advanced assimilation method. However, the average ensemble
spread for $MAAT_s$ in cold regions is still up to 45% greater than that of other regions. The relative $maxSWE_s$ among 4DVar
reanalyses is about 101% (56–186%), and is comparable to that derived from all five reanalyses.





## 3.2 Coincident low density of observations

We explored the density of in situ observations to contextualize variation in reanalysis performance. Reanalyses have the best agreement, or lowest $\mathrm{MAAT_s}$, in regions with rich observations, corresponding to the MAAT band from 0 to 10 °C, where 848 million (or about 10% of the global population) live (Fig. 1). In cold regions, the population is about 26 million (or 0.3%), with 80% in its warmest MAAT band of $-5$ to 0 °C. The density of in situ observations (0.08 station per $10^4$ km$^2$) is low compared to other regions, limiting the ability to constrain the reanalyses.

Interestingly, the $\mathrm{MAAT_s}$ in hot regions (MAAT $\geq 20$ °C) is about 0.4 °C (or 46%) lower compared to the extremely cold regions ($-20 \leq \mathrm{MAAT} \leq -10$ °C) although the density of observation in situ is comparably low (0.05 station per $10^4$ km$^2$). This indicates the prevalence of cryospheric phenomena such as glaciers, ice sheets, snow, and frozen soil (Fig. 1), and related physical processes pose important additional difficulties for the numerical weather prediction models (e.g., Cao et al., 2020; Domine et al., 2019). This is also visible in the 10-member ensemble of ERA5, taking into account random uncertainty in observations and parameterizations but having only a single prediction model and assimilation system (Fig. 1).

## 4 Implications

Reanalyses are produced by assimilating a broad range of historical observations into numerical weather prediction models. As such, they manifest how well we can estimate the state of the atmosphere and land surface globally, based on process knowledge and observation. They are widely considered a key data source for climate studies (Baatz et al., 2021) and the delivery of services (e.g., Dee et al., 2014; Gruber et al., 2023). This is especially true in cold regions, where we show that the quality of reanalyses in key variables is much lower than elsewhere.

While cold regions are remote and sparsely populated overall, this reduced quality of reanalyses matters. Environmental changes in cold regions involve feedbacks and tipping elements that affect Earth in its entirety. Obstacles in resolving the underlying processes likely point to similar gaps in knowledge and capabilities in the context of numerical weather prediction models. Furthermore, climate-driven changes in cold regions will be more profound than in many other terrestrial areas globally. As such, the ability to provide services and support climate research has an outsized importance for enabling resilient communities and resource extraction, as well as for national security and disaster preparedness.

*Data availability.* The mean annual air temperature and snow water equivalent datasets used here are publicly available on Zenodo (https://doi.org/10.5281/zenodo.14216654). The reanalyses are publicly available from their sponsoring agencies (Accessed on 24-12-2024). The JRA-3Q (https://search.diasjp.net/en/dataset/JRA3Q), and JRA-55 (https://search.diasjp.net/en/dataset/JRA55) are from Japan Meteorological Agency, ERA5 is from Climate Data Store (https://doi.org/10.24381/cds.f17050d7), MERRA-2 is from Goddard Earth Sciences Data and Information Services Center (https://doi.org/10.5067/0JRLVL8YV2Y4), and NCEP2 is from the NOAA Physical Sciences Laboratory (https://psl.noaa.gov/data/gridded/data.ncep.reanalysis2.html).



135 *Author contributions.* B.C. carried out this study by conducting simulations and analyses. S.G. proposed the initial idea and guided the study. Both authors wrote the text.

*Competing interests.* The contact author has declared that none of the authors has any competing interests.

*Acknowledgements.* This study was supported by the National Natural Science Foundation of China (grant no. 42422608), the Youth Innovation Promotion Association of the Chinese Academy of Sciences (grant no. 2023075) to B. Cao. S. Gruber is supported by NSERC
140 PermafrostNet (NETGP 523228-18) and NSERC (RGPIN-2020-04783).



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





**Figure 1.** The 1991–2020 average ensemble spread of (A) mean annual air temperature (MAAT) and (B) relative maximum snow water equivalent (MaxSWE) among different reanalyses. Land area and population are shown for context. Values are summarized in intervals of 5 °C for the ensemble mean of MAAT. Only reanalysis cells with a significant ($P < 0.05$) trends are used for the analysis of change. Blue numbers express low population counts in million. The occurrence of cryosphere elements, estimated as the probability of occurrence during the analysis period, is scaled per MAAT bin of 0.1 °C (see Methods). The red (3DVar and 4DVar) and green (4DVar only) lines represent ensembles of differing numerical weather prediction models and assimilation systems, whereas the yellow line (ERA5) represents uncertainty in observations and physical parameterizations in a single modelling and assimilation system. The peak in the trend of MaxSWE observed for MAAT class from −15 °C to −20 °C is caused by increased uncertainty in ice-free areas of Greenland and Antarctic.



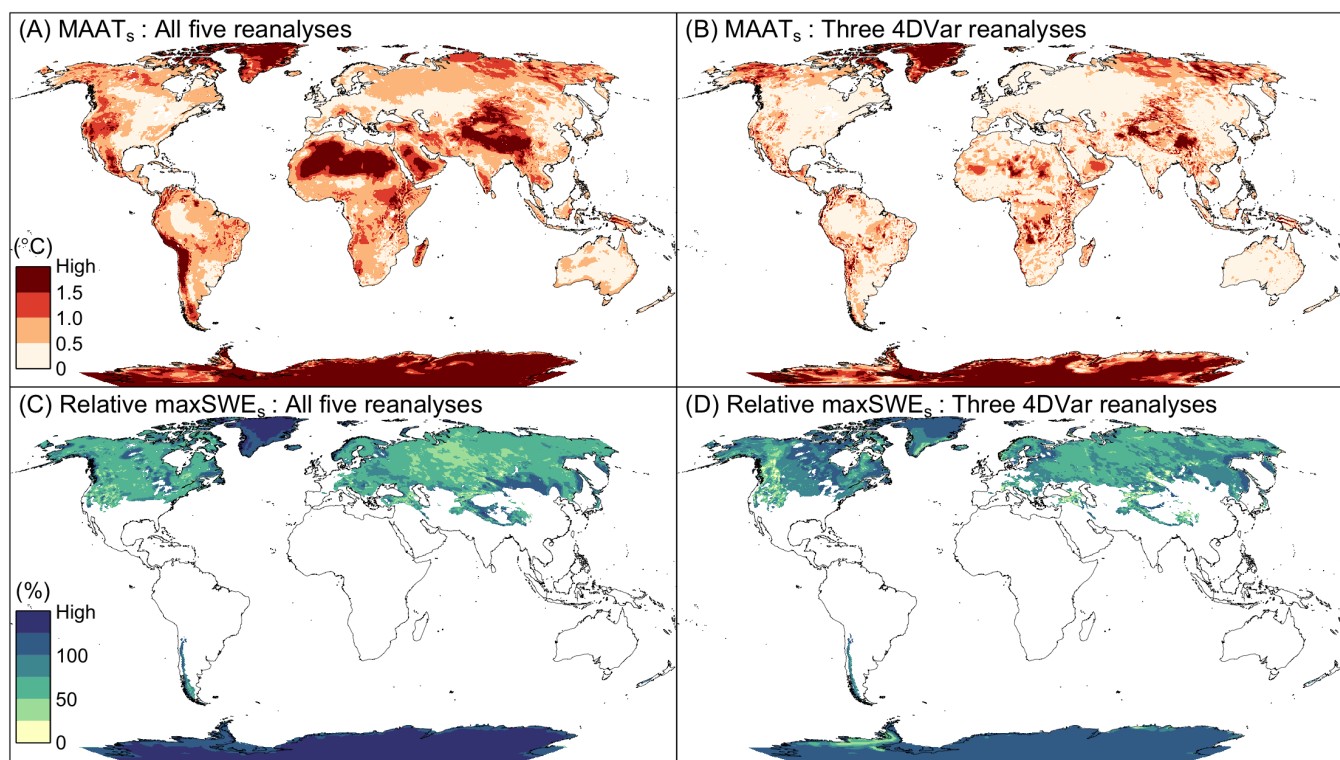

**Figure 2.** The 1991–2020 average ensemble spread of mean annual air temperature (MAAT$_s$) and relative spread of maximum snow water equivalent (maxSWE$_s$). Only areas with a mean maxSWE$_s$ greater than 0.0125 m (0.05 m snow height at a snow density is 250 kg m$^{-3}$) are shown. Snow water equivalent is not available for the two continental ice sheets in MERRA-2, and therefore, not included in these regions.