# Peer review of "Supporting Information for "Reanalyses underperform in cold regions, raising concerns for climate services and research""

_EGUsphere, 2025_

## Referee Comment (RC2)

Reviewer #2
**General Comments:**
This study by Cao and Gruber provides a timely evaluation of reanalysis performance in cold regions, underscoring key challenges in modeling mean annual air temperature (MAAT) and snow water equivalent (SWE). Presenting the work as a Brief Communication is appropriate, as it efficiently draws attention to issues of broad relevance. While a full uncertainty attribution is beyond the current scope—as noted in the response to Reviewer #1—the study lays a solid foundation for future work and enables rapid dissemination of important insights.

**Recommendation**: Minor revisions are recommended to improve clarity and strengthen the manuscript's impact. The core message is clear and timely; addressing the points below —primarily through brief clarifications or additions—would enhance the study's novelty and broader applicability without requiring new analyses.

**Additional Insights for Consideration:**
**1. Clarify the role of ERA5-ENS:** The use of the ERA5 ensemble is a valuable aspect of the study. However, the distinction between uncertainty within a single system (ERA5-ENS) and the broader inter-reanalysis spread could be further emphasized. This contrast may offer readers a clearer sense of where structural versus internal uncertainties dominate.

The notably smaller spread in ERA5-ENS (yellow line, Fig. 1) compared to the full multi-reanalysis ensemble warrants explicit discussion. This comparison could provide valuable insights into the relative importance of different uncertainty sources.

**2. Highlight the spatial dimension of spread:** The spatial maps in Figure 2 could be enhanced by including a difference panel or masking approach (e.g., isolating high-cryosphere, low-station-density zones). This could help illustrate where the reanalysis disagreement is most consequential for downstream applications.

Could the authors clarify whether the larger spread in high-altitude regions (e.g., High Asia, Greenland Ice Sheet) is primarily tied to low observation density or elevation-dependent processes (e.g., snow physics)? For instance, does Greenland's spread pattern align more with other high-altitude regions (suggesting elevation-driven uncertainty) or with low-altitude cold regions like Arctic tundra (suggesting temperature-driven uncertainty)? A brief discussion of how these factors interact in different cryospheric regimes would strengthen the interpretation of Figure 2.

**3. Cryospheric Regime Variability:** The study treats cold regions (MAAT < 0°C), yet these areas encompass diverse cryospheric regimes (e.g., high-altitude glaciers versus Arctic tundra). To enhance the analysis, the authors might consider: Stratifying results by cryosphere type: Does MAAT spread differ significantly between permafrost-dominated versus snow-dominated regions? For instance, snow-physics errors (such as MERRA-2's precipitation correction limitations) may be predominant in mountainous areas, while soil-thermal biases (Cao et al., 2020) might dominate in permafrost zones. A supplementary map or table categorizing spread according to cryosphere classification (using the already collected PZI and snow cover data) could reveal important process-specific discrepancies.

**4. Elevation Dependencies:** The Tibetan Plateau case (mentioned in the text) demonstrates notable elevation-dependent warming (Gao et al., 2018). Does MAAT spread show similar elevation dependence? Such analysis could help distinguish between:
a) Station-density effects (where lower elevations typically have better observational coverage)
b) Model physics limitations (e.g., inadequate representation of elevation-specific processes like katabatic winds or radiation biases)

**5. Temporal Analysis of Spread:** Given that Arctic amplification rates have shown distinct shifts approximately between 1990-2005 and 2015-2020 (Rantanen et al., 2022), it would be valuable to examine whether ensemble spread has correspondingly decreased in more recent years (particularly post-2000 with improved satellite assimilation). A persistent spread despite improved observations would strongly indicate fundamental process-representation issues beyond observational limitations.

**Suggested Improvements:**
1. The inclusion of a simple elevation-binned analysis (using existing data) could significantly strengthen the interpretation of regional variations in spread.
2. A brief discussion of how cryosphere type and elevation might interact to produce the observed spread patterns would enhance the manuscript's conceptual framework.
3. The temporal analysis of spread could be incorporated without requiring substantial new analysis, as the study already covers 1991-2020.

**Specific Comments:**
L36: Could the authors clarify what defines the 1991–2020 period as 'high quality' for reanalyses? For example, does this refer to improved satellite data assimilation, expanded observational networks, or advances in modeling? A brief explanation would help contextualize the results." Also, what do the authors mean by high quality observation in L56?
Lines 97-100: "Qin et al. (2020); Kraaijenbrink et al. (2021). ">> (Qian et al., 2020; Kraaijenbrink et al., 2021)
"Cao et al. (2020); Domine et al. (2019)" >> (Cao et al., 2020; Domine et al., 2019)
"Reichle et al. (2017)" >> (Reichle et al., 2017)

L159:(CDS), C.C.C.S.C.C.D.S? Could the authors update it to follow the recommended format from the CDS website:" Copernicus Climate Change Service, Climate Data Store, (2021): … (**Accessed on DD-MMM-YYYY**), 10.24381/cds.cf5f3bac"

Figure 1: The dotted line in the legend is difficult to associate with the corresponding dashed color lines in the figure. It took considerable effort to interpret their meaning correctly. I recommend clarifying the legend by explicitly stating that solid lines represent the mean state and dashed lines indicate the trend (e.g., "(solid: mean, dashed: trend, left Y-axis)" after each color label), while shaded areas on right Y-axis. This would significantly improve readability and interpretation.

**Reference:** Rantanen, M. et al. The Arctic has warmed nearly four times faster than the globe since 1979. *Commun. Earth Environ.* 3, 168 (2022).

---

## Author Comment (AC2)

**Author's Responses to Steven Margulis's comments on *"Brief communication: Reanalyses underperform in cold regions, raising concerns for climate services and research"**

Bin Cao[1], Stephan Gruber[2]

[1]State Key Laboratory of Tibetan Plateau Earth System Environment and Resources (TPESER), National Tibetan Plateau Data Center (TPDC), Institute of Tibetan Plateau Research, Chinese Academy of Sciences, Beijing, China
[2]Department of Geography & Environmental Studies, Carleton University, Ottawa, ON, Canada

**Correspondence**: Bin Cao (bin.cao@itpcas.ac.cn)

This is nice work, highlighting the issue of the high uncertainty of estimates of cold-land processes in reanalysis that are often used for making assessment of snow-derived water availability and how it may be changing. Some recent work that compared some of these global products to an observationally-constrained snow reanalysis dataset shed similar light and may be worth including in the Introduction for context.

Response: In a potential revision, we will incorporate Fang et al., (2023) in the Introduction, and Zhou et al., (2024) in Results (Sec. 3.1).:

*"Other studies report the performance of reanalyses for specific variables and places (e.g., Graham et al., 2019; Cao et al., 2020; **Fang et al., 2023**)."*

*"Previous studies reported that the snow uncertainties in mountains are related to the performance of numerical weather prediction models in representing precipitation and snow processes (Domine et al., 2019, Cao et al., 2020, **Zhou et al., 2024**), especially the well-known bias in MERRA-2 precipitation (Reichle et al., 2017)."*

**References**

Fang, Y., Y. Liu, D. Li, H. Sun, and S.A. Margulis, 2023. Spatiotemporal snow water storage uncertainty in the midlatitude American Cordillera, The Cryosphere, 17, 5175–5195, https://doi.org/10.5194/tc-17-5175-2023

Liu, Y., Y. Fang, D. Li, and S.A. Margulis, 2022. How well do global snow products characterize snow storage in High Mountain Asia? Geophysical Research Letters, 49, e2022GL100082. https://doi.org/10.1029/2022GL100082

---

## Author Comment (AC3)

**Author's Responses to RC1's comments on *"Brief communication: Reanalyses underperform in cold regions, raising concerns for climate services and research"**

Bin Cao[1], Stephan Gruber[2]

[1]State Key Laboratory of Tibetan Plateau Earth System Environment and Resources (TPESER), National Tibetan Plateau Data Center (TPDC), Institute of Tibetan Plateau Research, Chinese Academy of Sciences, Beijing, China
[2]Department of Geography & Environmental Studies, Carleton University, Ottawa, ON, Canada

**Correspondence**: Bin Cao (bin.cao@itpcas.ac.cn)

The authors would like to thank the reviewer for their constructive feedback, and the thorough assessment of the manuscript. Below, we provide a point-by-point response to each comment. Reviewer comments are given in black and responses in blue. Additionally, we have included details of how we intend to address these changes in a revised submission.

Cao and Gruber investigate the performance of five modern reanalyses (JRA-3Q, ERA5, MERRA-2, JRA-55 and NCEP2) over cold regions, with a focus on air temperature and snow water equivalent (SWE). They show that the ensemble spread in mean annual air temperature (MAAT) in reanalyses is up 90% greater over cold regions (defined as regions with a MAAT $< 0°C$) relative to regions with a MAAT $\geq 0°C$. The study explicitly shows the relationship between station density and ensemble spread for both SWE and MAAT and is able to show that the reduced reanalysis performance is at least partially related to the low station density over cold regions.

The bulk of the conclusions come from Figure 1, which show the average ensemble spread in MAAT (Panel A) and SWE (Panel B) binned by MAAT. The station density in each MAAT bin, and the proportion of the grid cells within the MAAT bin covered by ice sheets and glaciers, snow cover, permafrost, and seasonally frozen ground is also shown. Herrington et al. (2024) showed a similar plot for soil temperature, that identified the average reanalysis spread in soil temperatures binned by MAAT, against sample size, though it didn't explicitly consider station density, or the proportion of the grid cells covered by the cryosphere - so this is a novel analysis (along with the focus on SWE and MAAT).

While Figure 1 clearly shows a clear correlation between ensemble spread, station density, and the presence of cryospheric elements, there is no attempt to separate the contributions of low station density from those related to inadequate representation of cold region processes in reanalyses. As a reader it raises questions as to what the relative contributions of station density, and inadequate process representation to the ensemble spread in MAAT and SWE are?

To me, the novel science is in quantifying what proportion of the uncertainty can be attributed to station density, and what proportion is related to inadequate representation of cold region processes, which have been thoroughly discussed in the literature (e.g. Broxton et al., 2016; Cao et al., 2020, 2022; Hu et al., 2019; Mortimer et al., 2020; Wang et al., 2019). Thus, I recommend that the authors extend their analysis to explicitly quantify what proportion of the uncertainties or spread in MAAT and SWE can be attributed to the low station density in cold regions, and what proportion can be attributed to the inadequate process representation in the products. This will greatly enhance the contribution of the paper to the literature and provide the community with useful and quantifiable estimates of the uncertainty attribution.

Response (given in AC1 at `https://doi.org/10.5194/egusphere-2025-575-AC1`): This is an important point that has also been raised by colleagues who commented on an earlier version of the manuscript. We have made a deliberate decision here: A detailed and conclusive analysis of the causes for the large spread in cold regions will likely be an involved process requiring a broad range of knowledge, skills, and perspectives that differ from ours, and that will take time to bring together in a research project. We made a choice to expose our finding as a Brief Communication quickly to motivate and accelerate this research.

**Specific Comments**

- P2, L31: Why was NCEP2 investigated over NCEP CFSR/CFSv2, for example? NCEP CFSR/CFSv2 is available at a much higher resolution than NCEP2, and is available over the period of analysis (1991-2020).
  Response (given in AC1 at `https://doi.org/10.5194/egusphere-2025-575-AC1`): In selecting reanalyses to include, we have opted to not include CFSR/CFSv2 because it mixes two differing simulation and assimilation systems.

- P2, L36-L37: What do the authors mean by the statement "the decades 1991-2020 were used, likely a period of high

quality for reanalyses." Is there a particular standard by which the authors determined this? Some clarification would be helpful.

Response: In the revision, we will change this part to:

*"The three most recent decades 1991–2020, which had improved satellite observation and data assimilation (Hersbach et al., 2019), were used."*

In addition, we will 1) add a brief description regarding to the temporal changes of spread (see Sec. 3.1 Lower agreement among reanalyses in cold regions); and 2) revise Figure 2 by adding the temporal change of $MAAT_s$ and relative $maxSWE_s$.

*"The temporal analyses revealed that the spread in MAAT and maxSWE was generally reduced since 1980, with the increased assimilation of satellite datasets (Fig. 2E and F, Hersbach et al., 2019). For example, the $MAAT_s$ in 2010s was reduced by 0.23 °C compared to that in 1960s for the 4DVar reanalyses. But a persistent spread found since 1980 despite improved observations may indicate process-representation issues in the numerical weather prediction models.*

- P2, L38: What is the CDS? It appears that CDS is in brackets, but the acronym is not defined. I presume this may be the Climate Data Store?

  Response: The citation will be revised as below.

  Copernicus Climate Change Service, Climate Data Store, (2021): Global land surface atmospheric variables from 1755 to 2020 from comprehensive in-situ observations. Copernicus Climate Change Service (C3S) Climate Data Store (CDS). (Accessed on 21-04-2025), 10.24381/cds.cf5f3bac

- P4, L101: The all 5-reanalysis value for MAAT spread was (1.5°C, 0.5°C-3.0°C) - is this statistically different from the value for the 4DVar reanalyses reported here?

  Response: No. The MAAT (1.5, 0.5–3.0°C *vs.* 1.3, 0.3–2.9°C) as well as SWE (105, 51–206% *vs.* 101%, 56–186%) spread based on all 5-reanalysis and 4DVar reanalyses are generally close.

  *"Compared to all five reanalyses, the 4DVar reanalyses show a reduced spread in MAAT (1.3, 0.3–2.9 °C) and its trend (0.13, 0.04–0.24 °C dec$^{-1}$), as expected from a consistent and more advanced assimilation method. However, the average ensemble spread for $MAAT_s$ in cold regions is still up to 45% greater than that of other regions. The relative $maxSWE_s$ among 4DVar reanalyses is about 101% (56–186%), and is comparable to that derived from all five reanalyses."*

- P10, Figure 1: What do the dashed lines represent in Figure 1? I don't see a dashed line in the figure legend?

  Response: The dashed lines represent trend for MAAT and SWE. We will revise the figure and caption to clarify (see Fig. 1).

- P11, Figure 2: It may be helpful to have a "difference" panel between the All 5 reanalyses and the three 4DVar reanalyses to highlight the regions with the largest differences, particularly for SWE, since it is a little harder to notice the differences in Panel B.

  Response: We agree the "difference" between the all 5 reanalyses and the three 4DVar reanalyses would provide additional helpful information. We treat 4DVar reanalyses separately *"As better performance is expected from the newer 4DVAR reanalyses"* rather than demonstrating 4D. Our results, however, indicated the reanalyses significantly underperformed in cold regions regardless the assimilation method. In addition, 3DVar and 4DVar reanalyses generally show very close performance. To keep the manuscript "brief", the difference are given in Supporting Information (SI) as Figure S2 (see below).

**References**

Broxton, P. D., Zeng, X., and Dawson, N.: Why Do Global Reanalyses and Land Data Assimilation Products Underestimate Snow Water Equivalent?, J. Hydrometeorol., 17, 2743–2761, https://doi.org/10.1175/JHM-D-16-0056.1, 2016.

Cao, B., Gruber, S., Zheng, D., and Li, X.: The ERA5-Land soil temperature bias in permafrost regions, The Cryosphere, 14, 2581–2595, https://doi.org/10.5194/tc-14-2581-2020, 2020.

Cao, B., Arduini, G., and Zsoter, E.: Brief communication: Improving ERA5-Land soil temperature in permafrost regions using an optimized multi-layer snow scheme, The Cryosphere, 16, 2701–2708, https://doi.org/10.5194/tc-16-2701-2022, 2022.

Herrington, T. C., Fletcher, C. G., and Kropp, H.: Validation of Pan-Arctic Soil Temperatures in Modern Reanalysis and Data Assimilation Systems, The Cryosphere, 18, 1835–1861, https://doi.org/10.5194/tc-18-1835-2024, 2024.

Hersbach, H., Bell, W., Berrisford, P., Horányi, A., J, M.-S., Nicolas, J., Radu, R., Schepers, D., Simmons, A., Soci, C., Dee, D., 2019. Global reanalysis: goodbye ERA-Interim, hello ERA5. ECMWF Newsletter. https://doi.org/10.21957/vf291hehd7.

Hu, G., Zhao, L., Li, R., Wu, X., Wu, T., Xie, C., Zhu, X., and Su, Y.: Variations in soil temperature from 1980 to 2015 in permafrost regions on the Qinghai-Tibetan Plateau based on observed and reanalysis products, Geoderma, 337, 893–905, https://doi.org/10.1016/j.geoderma.2018.10.044, 2019.

Mortimer, C., Mudryk, L., Derksen, C., Luojus, K., Brown, R., Kelly, R., and Tedesco, M.: Evaluation of long-term Northern Hemisphere snow water equivalent products, The Cryosphere, 14, 1579–1594, https://doi.org/10.5194/tc-14-1579-2020, 2020.

Wang, C., Graham, R. M., Wang, K., Gerland, S., and Granskog, M. A.: Comparison of ERA5 and ERA-Interim near-surface air temperature, snowfall and precipitation over Arctic sea ice: effects on sea ice thermodynamics and evolution, The Cryosphere, 13, 1661–1679, https://doi.org/10.5194/tc-13-1661-2019, 2019.

[Figure]

Figure 1: The 1991–2020 average ensemble spread of (A) mean annual air temperature (MAAT) and (B) relative maximum snow water equivalent (MaxSWE) among different reanalyses. The red (3DVar and 4DVar) and green (4DVar only) lines represent ensembles of differing numerical weather prediction models and assimilation systems, whereas the yellow line (ERA5) represents uncertainty in observations and physical parameterizations in a single modelling and assimilation system. **The solid lines represent the mean state and dashed lines indicate the trend (left vertical axis).** Land area and population are shown for context **(right vertical axis)**. Values are summarized in intervals of 5 °C for the ensemble mean of MAAT. The occurrence of cryosphere elements, estimated as the probability of occurrence during the analysis period, is scaled per MAAT bin of 0.1 °C (see Methods). Only reanalysis cells with a significant ($P < 0.05$) trends are used for the analysis of change. Blue numbers express low population counts in million. The peak in the trend of MaxSWE observed for MAAT class from −15 °C to −20 °C is caused by increased uncertainty in ice-free areas of Greenland and Antarctic.

[Figure]

Figure 2: The 1991–2020 average ensemble spread of mean annual air temperature (MAAT$_s$) and relative spread of maximum snow water equivalent (maxSWE$_s$). Only areas with a mean maxSWE$_s$ greater than 0.0125 m (0.05 m snow height at a snow density is 250 kg m$^{-3}$) are shown. Snow water equivalent is not available for the two continental ice sheets in MERRA-2, and therefore, MERRA-2 is not included in these regions. The overall temporal changes for (E) MAAT$_s$ and (F) relative maxSWE$_s$ was derived with the reference period of 2011–2020, and a positive value means the spread is reduced relative to the referenced period. In E and F, the soil lines represent the mean state and shaded areas indicate 10th to 90th percentile. The spread difference between all five reanalyses and three 4DVar reanalyses is given in Fig. S2.

Figure S2: The difference between all five reanalyses and three 4DVar reanalyses for (A) mean annual air temperature (B) and relative maximum snow water equivalent (maxSWE). Blue areas indicate in improvement (smaller spread) in the 4DVar results relative to all five reanalyses, Red areas indicate a deterioration (greater spread) in the 4DVar results relative to all five reanalyses.

---

## Author Comment (AC4)

**Author's Responses to RC2's comments on *"Brief communication: Reanalyses underperform in cold regions, raising concerns for climate services and research"**

Bin Cao[1], Stephan Gruber [2]

[1]State Key Laboratory of Tibetan Plateau Earth System Environment and Resources (TPESER), National Tibetan Plateau Data Center (TPDC), Institute of Tibetan Plateau Research, Chinese Academy of Sciences, Beijing, China
[2]Department of Geography & Environmental Studies, Carleton University, Ottawa, ON, Canada

**Correspondence**: Bin Cao (bin.cao@itpcas.ac.cn)

*The authors would like to thank the reviewer for their constructive feedback and thorough assessment of our manuscript. Below, we provide a point-by-point response to each comment, reviewer comments are given in* black, *responses are given in blue. Additionally, we have included details of how we intend to address these changes in a potential revised submission. Revised figure/table are presented at the end of our responses.*

This study by Cao and Gruber provides a timely evaluation of reanalysis performance in cold regions, underscoring key challenges in modeling mean annual air temperature (MAAT) and snow water equivalent (SWE). Presenting the work as a Brief Communication is appropriate, as it efficiently draws attention to issues of broad relevance. While a full uncertainty attribution is beyond the current scope–as noted in the response to Reviewer #1–the study lays a solid foundation for future work and enables rapid dissemination of important insights.

**Recommendation:** Minor revisions are recommended to improve clarity and strengthen the manuscript's impact. The core message is clear and timely; addressing the points below–primarily through brief clarifications or additions–would enhance the study's novelty and broader applicability without requiring new analyses.

**Additional Insights for Consideration**

**1. Clarify the role of ERA5-ENS:** The use of the ERA5 ensemble is a valuable aspect of the study. However, the distinction between uncertainty within a single system (ERA5-ENS) and the broader inter-reanalysis spread could be further emphasized. This contrast may offer readers a clearer sense of where structural versus internal uncertainties dominate.

The notably smaller spread in ERA5-ENS (yellow line, Fig. 1) compared to the full multi-reanalysis ensemble warrants explicit discussion. This comparison could provide valuable insights into the relative importance of different uncertainty sources.
Response:
**In Methods, Sec. 2.1**, this part will be revised as below to clarify: *"The 10-member ensemble of ERA5, which quantifies uncertainties in the ERA5 assimilation and modeling system, was also included here for comparison. The ERA5 ensemble provides an opportunity to show how parameter uncertainty in one reanalysis system compares with the spread between structurally different reanalyses."*

**In Results, Sec. 3.1**, we will add below part to clarify. *"Compared to the spread of multiple reanalyses that also differ in structure, the spread of the ERA5 ensemble, within a consistent assimilation system and representing mostly parametric uncertainty, is notably smaller, as excepted, i.e., 0.1 °C (0.0–0.3) for MAAT$_s$ and 1.0% (0–2.6) for relative maxSWE$_s$."*

**2. Highlight the spatial dimension of spread:** The spatial maps in Figure 2 could be enhanced by including a difference panel or masking approach (e.g., isolating highcryosphere, low-station-density zones). This could help illustrate where the reanalysis disagreement is most consequential for downstream applications.

Could the authors clarify whether the larger spread in high-altitude regions (e.g., High Asia, Greenland Ice Sheet) is primarily tied to low observation density or elevation-dependent processes (e.g., snow physics)? For instance, does Greenland's spread pattern align more with other high-altitude regions (suggesting elevation-driven uncertainty) or with low-altitude cold regions like Arctic tundra (suggesting temperature-driven uncertainty)? A brief discussion of how these factors interact in different cryospheric regimes would strengthen the interpretation of Figure 2.

**4. Elevation Dependencies:** The Tibetan Plateau case (mentioned in the text) demonstrates notable elevation-dependent warming (Gao et al., 2018). Does MAAT spread show similar elevation dependence? Such analysis could help distinguish between:

a) Station-density effects (where lower elevations typically have better observational coverage)

b) Model physics limitations (e.g., inadequate representation of elevation-specific processes like katabatic winds or radiation biases)

Response: These are the responses to the Additional Insights for Consideration 2 & 4.

In the Introduction, we intended to clarify that the Arctic amplification warming and elevation-dependent warming remain controversial depending on the datasets used, rather than demonstrating unambiguously that elevation-dependent warming occurs uniformly.

We agree the spatial and elevation-dependent analyses can provide additional insights. However, the coarse spatial resolution of reanalysis leads to inadequate representation of elevation-specific processes like precipitation, katabatic winds, or radiation in the numerical weather prediction models used.

Instead, we intend to use terrain ruggedness in the revision to distinguish the added uncertainties due to complex terrain. The roughness was estimated following Gruber (2012), and is a proxy for the potential uncertainties arising from scale effects related to the coarse reanalysis grid.

In Sec. 3, we will add: *"Figure S3 shows how ensemble spread increases with terrain ruggedness. On the other hand, the spread on the two continental ice sheets is also high even though the terrain is flat. This is likely because of inadequate representation of processes involving ice, snow, and firn."*

In the supporting information, we will add the method for roughness estimation.

*Text S1. Terrain Ruggedness*
*The terrain ruggedness (rug, m km$^{-1}$) is derived largely following Gruber 2012.*

$$rug = \frac{E_{std}}{\sqrt{A}} \tag{1}$$

*where $E_{std}$ (m) and A (km$^{-2}$) is the elevation standard and area for a analysis grid of 0.25°. The elevation is from GTOPO30 with a spatial resolution of 30 arc-second or ~1 km.*

**3. Cryospheric Regime Variability:** The study treats cold regions (MAAT $< 0°$C), yet these areas encompass diverse cryospheric regimes (e.g., high-altitude glaciers versus Arctic tundra). To enhance the analysis, the authors might consider: Stratifying results by cryosphere type: Does MAAT spread differ significantly between permafrost-dominated versus snow-dominated regions? For instance, snow-physics errors (such as MERRA-2's precipitation correction limitations) may be predominant in mountainous areas, while soil-thermal biases (Cao et al., 2020) might dominate in permafrost zones. A supplementary map or table categorizing spread according to cryosphere classification (using the already collected PZI and snow cover data) could reveal important process-specific discrepancies.

Response: In the revision, the spread will be aggregated for each specific cryosphere element, including: ice sheets and glaciers, snow cover, permafrost, and seasonally frozen ground, following Figure 2A. To achieve this, we will add a new table (Table 1) with spread for each specific element. On the other hand, we will move the original Table 1 of reanalysis information to supporting information (as Table S1) in order to avoid excessive length in the manuscript. These four elements may have some overlap in spatial due the corse spatial resolution of reanalysis and inherent features of the cryosphere. For these reasons, the classification is carefully treated. In sec. 2.5 Cryosphere occurrence of the revision, we will clarify as below.

*"To distinguish the variability between cryospheric regimes, the spread was individually analyzed for each cryosphere element. Ice sheets-dominated areas include the Greenland and Antarctic ice sheets; areas with more than 30 snow-covered days were considered snow-dominated; areas with PZI $\geq$ 0.1 were considered permafrost-dominated area; and areas with frozen soil $\geq$ 30 days (excluding permafrost-dominated areas) were considered seasonally-frozen-ground-dominated areas."*

In Sec. 3.1, we will add a brief clarification for specific cryopshere element variability. *" While the overall spread is remarkable in cold regions, the ice sheet areas show most significant spread, which is about 2.3 times greater for MAAT and 1.7 times greater for maxSWE compared to that in seasonally-frozen-ground-dominated areas based on all five reanalyses (Table 1, Fig. 1)."*

**5. Temporal Analysis of Spread:** Given that Arctic amplification rates have shown distinct shifts approximately between 1990-2005 and 2015-2020 (Rantanen et al., 2022), it would be valuable to examine whether ensemble spread has corre-

Table 1: The spread of mean annual air temperature (MAAT$_s$, °C) and relative maximum snow water equivalent (maxSWE$_s$, %) for the areas occupied by specific cryosphere elements.

| Cryosphere element | MAAT$_s^{all}$ | MAAT$_s^{4DV}$ | SWE$_s^{all}$ | SWE$_s^{4DV}$ |
|---|---|---|---|---|
| Ice sheets & glaciers | 2.27 (1.05–3.58) | 2.03 (0.62–3.79) | 197.3 (171.6–206.6) | 154.0 (92.7–190.2) |
| Snow cover | 0.77 (0.28–1.39) | 0.52 (0.10–1.15) | 79.9 (52.9–115.0) | 72.3 (49.6–104.5) |
| Permafrost | 0.97 (0.47–1.65) | 0.80 (0.24–1.49) | 74.6 (50.4–108.4) | 82.7 (53.7-123.1) |
| Seasonally frozen ground | 0.68 (0.24–1.29) | 0.37 (0.08–0.78) | 72.0 (49.1–105.9) | 78.7 (52.4–108.4) |

Values are reported as mean (10th to 90th percentile).
Superscripts distinguish all five (all) reanalyses and the three 4DVar modern reanalyses (4DV), only.

spondingly decreased in more recent years (particularly post-2000 with improved satellite assimilation). A persistent spread despite improved observations would strongly indicate fundamental process-representation issues beyond observational limitations.

Regarding to the temporal changes of spread, we will extend the analysis back to the earliest period for which data is available. We will add a brief description (Sec. 3.1) and two sub-figure in Figure 2 (E and F).

In Results Sec. 3.1, we will add *"The temporal analyses revealed that the spread in MAAT and maxSWE was generally reduced since 1980, with the increased assimilation of satellite datasets (Fig. 2E and F, Hersbach et al., 2019). For example, the MAAT$_s$ in 2010s was reduced by 0.23 °C compared to that in 1960s for the 4DVar reanalyses. But a persistent spread found since 1980 despite improved observations may indicate process-representation issues in the numerical weather prediction models."*

**Suggested Improvements:**
1. The inclusion of a simple elevation-binned analysis (using existing data) could significantly strengthen the interpretation of regional variations in spread.
Response: Please see our responses to the Additional Insights for Consideration 4 Elevation Dependencies.

2. A brief discussion of how cryosphere type and elevation might interact to produce the observed spread patterns would enhance the manuscript's conceptual framework.
Response: Please see our responses to the Additional Insights for Consideration 3 and 4.

3. The temporal analysis of spread could be incorporated without requiring substantial new analysis, as the study already covers 1991-2020.
Response: Please see our responses to the Additional Insights for Consideration 5 Temporal Analysis of Spread.

**Specific Comments:**

L36: Could the authors clarify what defines the 1991–2020 period as 'high quality' for reanalyses? For example, does this refer to improved satellite data assimilation, expanded observational networks, or advances in modeling? A brief explanation would help contextualize the results." Also, what do the authors mean by high quality observation in L56?
Response: We will revise as below to clarify

*"The three most recent decades 1991–2020, which had improved satellite observation and data assimilation, were used."*

Lines 97-100: "Qin et al. (2020); Kraaijenbrink et al. (2021)." >> (Qian et al., 2020; Kraaijenbrink et al., 2021) "Cao et al. (2020); Domine et al. (2019)" >> (Cao et al., 2020; Domine et al., 2019) "Reichle et al. (2017)" >> (Reichle et al., 2017)
Response: Will be revised.

L159:(CDS), C.C.C.S.C.C.D.S? Could the authors update it to follow the recommended format from the CDS website: "Copernicus Climate Change Service, Climate Data Store, (2021): (Accessed on DD-MMM-YYYY), 10.24381/cds.cf5f3bac"
Response: Will be revised as below.

Copernicus Climate Change Service, Climate Data Store, (2021): Global land surface atmospheric variables from 1755 to 2020 from comprehensive in-situ observations. Copernicus Climate Change Service (C3S) Climate Data Store (CDS). (Accessed on 21-04-2025), 10.24381/cds.cf5f3bac

Figure 1: The dotted line in the legend is difficult to associate with the corresponding dashed color lines in the figure. It took considerable effort to interpret their meaning correctly. I recommend clarifying the legend by explicitly stating that solid lines represent the mean state and dashed lines indicate the trend (e.g., "(solid: mean, dashed: trend, left Y-axis)" after each color label), while shaded areas on right Y-axis. This would significantly improve readability and interpretation.

Response: We will revise the figure and caption to clarify.

Figure Caption: The 1991–2020 average ensemble spread of (A) mean annual air temperature (MAAT) and (B) relative maximum snow water equivalent (MaxSWE) among different reanalyses. The red (3DVar and 4DVar) and green (4DVar only) lines represent ensembles of differing numerical weather prediction models and assimilation systems, whereas the yellow line (ERA5) represents uncertainty in observations and physical parameterizations in a single modelling and assimilation system. **The solid lines represent the mean state and dashed lines indicate the trend (left vertical axis)**. Land area and population are shown for context **(right vertical axis)**. Values are summarized in intervals of 5 °C for the ensemble mean of MAAT. The occurrence of cryosphere elements, estimated as the probability of occurrence during the analysis period, is scaled per MAAT bin of 0.1 °C (see Methods). Only reanalysis cells with a significant ($P < 0.05$) trends are used for the analysis of change. Blue numbers express low population counts in million. The peak in the trend of MaxSWE observed for MAAT class from −15 °C to −20 °C is caused by increased uncertainty in ice-free areas of Greenland and Antarctic.

**References**

Rantanen, M. et al. The Arctic has warmed nearly four times faster than the globe since 1979. Commun. Earth Environ. 3, 168 (2022).

Gruber, S. Derivation and analysis of a high-resolution estimate of global permafrost zonation. The Cryosphere 6, 1 (2012), 221-233.

[Figure]

Figure 1: The 1991–2020 average ensemble spread of (A) mean annual air temperature (MAAT) and (B) relative maximum snow water equivalent (MaxSWE) among different reanalyses. The red (3DVar and 4DVar) and green (4DVar only) lines represent ensembles of differing numerical weather prediction models and assimilation systems, whereas the yellow line (ERA5) represents uncertainty in observations and physical parameterizations in a single modelling and assimilation system. **The solid lines represent the mean state and dashed lines indicate the trend (left vertical axis).** Land area and population are shown for context **(right vertical axis)**. Values are summarized in intervals of 5 °C for the ensemble mean of MAAT. The occurrence of cryosphere elements, estimated as the probability of occurrence during the analysis period, is scaled per MAAT bin of 0.1 °C (see Methods). Only reanalysis cells with a significant ($P < 0.05$) trends are used for the analysis of change. Blue numbers express low population counts in million. The peak in the trend of MaxSWE observed for MAAT class from −15 °C to −20 °C is caused by increased uncertainty in ice-free areas of Greenland and Antarctic.

[Figure]

Figure 2: The 1991–2020 average ensemble spread of mean annual air temperature ($MAAT_s$) and relative spread of maximum snow water equivalent ($maxSWE_s$). Only areas with a mean $maxSWE_s$ greater than 0.0125 m (0.05 m snow height at a snow density is 250 kg m$^{-3}$) are shown. Snow water equivalent is not available for the two continental ice sheets in MERRA-2, and therefore, not included in these regions. The overall temporal changes for (E) $MAAT_s$ and (F) relative $maxSWE_s$ was derived with the reference period of 2011–2020, and a positive value means the spread is reduced relative to the referenced period. In E and F, the solid lines represent the mean state and shaded areas indicate 10th to 90th percentile. The difference between all five reanalyses and three 4DVar reanalyses is given in Fig. S2.

[Figure]

Figure S3: The changes of ensemble spread for (A) mean annual air temperature (MAAT) and (B) relative maximum snow water equivalent (maxSWE) as a function of (C) terrain ruggedness. The points represent the mean spread and lines indicate 10th to 90th percentile.

---

## Referee Report (RR1)

**General Comments**

Cao and Gruber investigate the performance of five modern reanalyses (JRA-3Q, ERA5, MERRA-2, JRA-55 and NCEP2) over cold regions, with a focus on air temperature and snow water equivalent (SWE). The manuscript has been revised and addresses many of the comments that myself and the other referee had, resulting in a more robust analysis of the deficiencies of reanalyses over cold regions. I recommend that the manuscript be published following a few minor changes listed below.

**Specific Comments**

P1, L22: Why are reanalyses of higher importance over cold regions? It would be helpful to make the connection to spatial and temporal gaps in the observational record here.

Table 1, Page 4: Could the authors include the values of the spread for each variable over the study region as a whole? This would provide a strong visual contrast between performance over cold regions, and performance elsewhere.

P5, L110-114: If the difference between the 4DVar and all 5 reanalyses is not statistically significant, I would argue that this suggests that they are comparable; similar to what is mentioned for maxSWE. I suggest that the authors mention that the spread of the MAAT and relative maxSWE for the 4DVar is similar to that of all 5 products. Instead, focus on the main differences (i.e. that the spread for MAAT is up to 45% larger over cold regions, and that parametric uncertainty is an order of magnitude smaller than the structural uncertainty, etc.

P6, L145-147: The authors mention the importance of cold regions to understanding how the climate system responds to future changes. Do the authors have any suggestions for future research on the most critical changes that could be made to address the degraded performance over cold regions? I feel like this aspect is missing from the implications section.

**Technical Comments**

P6, L152: Replace "ERA5 is from Climate Data Store" with "ERA5 is from *the* Climate Data Store"

---

## Author Response (AR2)

**Author's Responses to comments on *"Brief communication: Reanalyses underperform in cold regions, raising concerns for climate services and research"**

Bin Cao[1], Stephan Gruber[2]

[1]State Key Laboratory of Tibetan Plateau Earth System Environment and Resources (TPESER), National Tibetan Plateau Data Center (TPDC), Institute of Tibetan Plateau Research, Chinese Academy of Sciences, Beijing, China
[2]Department of Geography & Environmental Studies, Carleton University, Ottawa, ON, Canada

**Correspondence**: Bin Cao (bin.cao@itpcas.ac.cn)

The authors would like to thank the reviewer for their constructive feedback and thorough assessment of our manuscript. Below, we provide a point-by-point response to each comment, reviewer comments are given in black, responses are given in blue. Additionally, we have included details of how we addressed these changes in a potential revised submission. Revised figure/table are presented at the end of our responses.

**Responses to RC1**

Cao and Gruber investigate the performance of five modern reanalyses (JRA-3Q, ERA5, MERRA-2, JRA-55 and NCEP2) over cold regions, with a focus on air temperature and snow water equivalent (SWE). The manuscript has been revised and addresses many of the comments that myself and the other referee had, resulting in a more robust analysis of the deficiencies of reanalyses over cold regions. I recommend that the manuscript be published following a few minor changes listed below.

**Specific Comments**

- P1, L22: Why are reanalyses of higher importance over cold regions? It would be helpful to make the connection to spatial and temporal gaps in the observational record here.
  Response: We reworked Paragraph 2 & 3 to clarify:

  *Understanding cold regions is important for informing local climate-change adaptation and climate action globally. Their climate conditions and dynamics, however, can be subject to disagreement. For example, previous studies suggested that the climate signal in cold regions could be different depending on the datasets used (Huang et al., 2017, Wang et al., 2017) and concluded that the 'warming hiatus' in the Arctic may be an artifact. Other studies report the performance of reanalyses for specific variables and places (e.g., Graham et al., 2019, Cao et al., 2020, Fang et al., 2023, Lan et al., 2025). This is because many cold-region processes react nonlinearly to changes near $0\,°C$ due to the ice-water phase transition, their analysis and simulation are extra sensitive to errors. In addition to these challenges, sparse in-situ observations increase the need and importance for atmospheric reanalyses as a tool for supporting climate research and services. While the quality of reanalyses is expected to be lower in cold regions, their quality is also less well known than elsewhere.*

- Table 1, Page 4: Could the authors include the values of the spread for each variable over the study region as a whole? This would provide a strong visual contrast between performance over cold regions, and performance elsewhere.
  Response: the overall spread for the cold and non-cold regions were added to Table 1. In addition, the MAAT spread are revised to be one-decimal, and $SWE_{max}$ to be integer.

- P5, L110-114: If the difference between the 4DVar and all 5 reanalyses is not statistically significant, I would argue that this suggests that they are comparable; similar to what is mentioned for maxSWE. I suggest that the authors mention that the spread of the MAAT and relative maxSWE for the 4DVar is similar to that of all 5 products. Instead, focus on the main differences (i.e. that the spread for MAAT is up to 45% larger over cold regions, and that parametric uncertainty is an order of magnitude smaller than the structural uncertainty, etc.
  Response: we revised as below:

*Compared to all five reanalyses, the MAAT$_s$ spread (1.3, 0.3–2.9 °C) and its trend (0.13, 0.04–0.24 °C dec$^{-1}$) for the 4DVar reanalyses are still significant in cold regions, and the average ensemble spread is up to 45% greater than that of other regions (Table 1, Figure 1 and 2), indicating the inherent issues regarding to the complex ice-related processes.*

- P6, L145-147: The authors mention the importance of cold regions to understanding how the climate system responds to future changes. Do the authors have any suggestions for future research on the most critical changes that could be made to address the degraded performance over cold regions? I feel like this aspect is missing from the implications section.
  Response: we agree. The suggestions for future research are added at the end of the manuscript (Sec. 4 Implications).

*We hope our analysis will help raise the awareness of how important cold-regions processes may be for NWP and reanalyses, and thus encourage greater focus on studies of individual cryosphere elements (e.g., Cao et al., 2022, Meloche et al., 2022) to inform research leading to future improvements in NWP.*

**Technical Comments**

- P6, L152: Replace "ERA5 is from Climate Data Store" with "ERA5 is from the Climate Data Store"
  Responses: Revised.

**References**

Cao, B., Arduini, G., and Zsoter, E.: Brief communication: Improving ERA5-Land soil temperature in permafrost regions using an optimized multi-layer snow scheme, The Cryosphere, 16, 2701–2708, https://doi.org/10.5194/tc-16-2701-2022, 2022.

Cao, B., Gruber, S., Zheng, D., and Li, X.: The ERA5-Land soil temperature bias in permafrost regions, The Cryosphere, 14, 2581–2595, https://doi.org/10.5194/tc-14-2581-2020, 2020.

Fang, Y., Liu, Y., Li, D., Sun, H., and Margulis, S. A.: Spatiotemporal snow water storage uncertainty in the midlatitude American Cordillera, The Cryosphere, 17, 5175–5195, https://doi.org/10.5194/tc-17-5175-2023, 2023.

Graham, R. M., Hudson, S. R., and Maturilli, M.: Improved performance of ERA5 in Arctic gateway relative to four global atmospheric reanalyses, Geophysical Research Letters, 46, 6138–6147, 2019.

Lan, S., Cao, B., Li, X., Sun, W., Wang, S., Ma, R., Sun, Z., and Guo, X.: Improved JRA-3Q Soil Temperature in Permafrost Regions, Journal of Climate, 38, 1611–1625, https://doi.org/10.1175/JCLI-D-24-0267.1, 2025.

Meloche, J., Langlois, A., Rutter, N., Royer, A., King, J., Walker, B., Marsh, P., and Wilcox, E. J.: Characterizing tundra snow sub-pixel variability to improve brightness temperature estimation in satellite SWE retrievals, The Cryosphere, 16, 87–101, https://doi.org/10.5194/tc-16-87-2022, 2022.

Table 1: The spread of mean annual air temperature (MAAT$_s$, °C) and relative maximum snow water equivalent (maxSWE$_s$, %) for the areas occupied by specific cryosphere elements.

| Cryosphere element | | MAAT$_s^{all}$ | MAAT$_s^{4DV}$ | SWE$_s^{all}$ | SWE$_s^{4DV}$ |
|---|---|---|---|---|---|
| Overall | Cold regions | 1.5 (0.5–3.0) | 1.3 (0.3–2.9) | 105 (51–206) | 101 (56–186) |
| | Non-cold regions | 0.8 (0.3–1.5) | 0.5 (0.1–0.9) | – | – |
| Ice sheets & glacier | | 2.3 (1.1–3.6) | 2.0 (0.6–3.8) | 197 (172–207) | 154 (93–190) |
| Snow cover | | 0.8 (0.3–1.4) | 0.5 (0.1–1.2) | 80 (53–115) | 72 (50–105) |
| Permafrost | | 1.0 (0.5–1.7) | 0.8 (0.2–1.5) | 75 (50–108) | 83 (54–123) |
| Seasonally frozen ground | | 0.7 (0.2–1.3) | 0.4 (0.1–0.8) | 72 (49–106) | 79 (52–108) |

Values are reported as mean (10th to 90th percentile).
Superscripts distinguish all five (all) reanalyses and the three 4DVar modern reanalyses (4DV), only.